# Information Needs and Communication Strategies for People with Coronary Heart Disease: A Scoping Review

**DOI:** 10.3390/ijerph20031723

**Published:** 2023-01-17

**Authors:** Clara C. Zwack, Carlie Smith, Vanessa Poulsen, Natalie Raffoul, Julie Redfern

**Affiliations:** 1School of Health Sciences, Faculty of Medicine and Health, University of Sydney, Sydney, NSW 2006, Australia; 2National Heart Foundation of Australia, Brisbane, QLD 4006, Australia; 3National Heart Foundation of Australia, Adelaide, SA 5000, Australia; 4National Heart Foundation Australia, Sydney, NSW 2011, Australia

**Keywords:** information, resources, coronary heart disease, digital health, education, cardiac rehabilitation, secondary prevention, text message, sensors, cardiovascular risk

## Abstract

A critical aspect of coronary heart disease (CHD) care and secondary prevention is ensuring patients have access to evidence-based information. The purpose of this review is to summarise the guiding principles, content, context and timing of information and education that is beneficial for supporting people with CHD and potential communication strategies, including digital interventions. We conducted a scoping review involving a search of four databases (Web of Science, PubMed, CINAHL, Medline) for articles published from January 2000 to August 2022. Literature was identified through title and abstract screening by expert reviewers. Evidence was synthesised according to the review aims. Results demonstrated that information-sharing, decision-making, goal-setting, positivity and practicality are important aspects of secondary prevention and should be patient-centred and evidenced based with consideration of patient need and preference. Initiation and duration of education is highly variable between and within people, hence communication and support should be regular and ongoing. In conclusion, text messaging programs, smartphone applications and wearable devices are examples of digital health strategies that facilitate education and support for patients with heart disease. There is no one size fits all approach that suits all patients at all stages, hence flexibility and a suite of resources and strategies is optimal.

## 1. Introduction

Cardiovascular disease (CVD), including coronary heart disease (CHD) and stroke, is the leading cause of death and disease burden globally [1]. CVD was responsible for approximately 32% of deaths in 2019 and the Global Burden of Disease data revealed that ischaemic heart disease affects approximately 126 million people [2]. Importantly, approximately a third of these occur in people who have prior disease and are therefore largely preventable [3,4]. The recent data from the SNAPSHOT ACS follow-up study (Australian patients) found that almost 20% of patients admitted to hospital with acute coronary syndrome (ACS) die within three years of discharge and 40% experience another hospitalisation for CVD in the same period [5]. With an aging population, more people surviving initial heart events, and an epidemic of lifestyle-related risk factors, the health burden is escalating globally [6]. Thus, improving post-discharge care through adherence to best practice secondary prevention strategies (healthy living, medicines and psychosocial support) is a current international priority [6,7].

Cardiac rehabilitation offers a traditional approach for providing structured secondary prevention. It is defined by the World Health Organisation as “the coordinated sum of activities required to influence favourably the underlying cause of CVD, as well as to provide the best possible physical, mental and social conditions, so that the patients may, by their own efforts, preserve or resume optimal functioning in their community and through improved health behaviour, slow or reverse progression of disease” [8]. Traditionally, programs have been conducted as a 6–10-week program of group-based exercise and education that is commenced within the initial six weeks of an event [9]. Cardiac rehabilitation has been shown to reduce morbidity and improve quality of life [10] and therefore, referral, participation and completion are recommended in Australian [11] and international [12,13] guidelines. However, research consistently shows unacceptably poor rates of referral (30% of those eligible), attendance (9% of eligible) and completion (<5% of eligible) [14]. Reasons are well documented and include transport, work/social commitments and lack of perceived need [15]. Further, adherence to evidence-based medications and lifestyle change typically start to decline within the first six months [16,17] and certain groups are less likely to attend structured programs, including women, those from culturally/linguistically diverse or low socioeconomic backgrounds [18].

From a systems perspective, the practicality and cost of providing traditional programs to all who are eligible (>60,000 new patients annually) across diverse geographical areas with cultural and linguistic diversity is a major barrier [19]. Therefore, there is increasing support for programs to be more flexible, culturally appropriate, multifaceted and integrated with patients’ primary health care providers to achieve optimal and sustainable benefits for most patients [20]. Recent advancements in technology and digitalisation of healthcare now provide opportunities to support patients in different ways. Digital health has the potential to close the gap of access inequities by improving the reach and strength of health systems [21]. This has become particularly relevant during the current COVID-19 pandemic in which social isolation has propelled the need for rapid proliferation of remote healthcare delivery.

A critical aspect of recovery and secondary prevention is ensuring patients have access to evidence-based information and acquire knowledge related to their condition. With adequate and appropriate information, patients and relevant carers are better placed to make decisions and implement optimal behavioural changes [22]. Ideally, information-sharing, decision-making and goal-setting should be patient-centred and evidenced based with consideration of patient need and preference [23,24]. There is a plethora of evidence highlighting the importance of information and education throughout the entire health journey, whether it be in the preventive stage, during treatment, in early convalescence, or longer-term rehabilitation [25]. However, the provision of information should ideally commence early after an acute event to increase confidence in managing symptoms and to improve the likelihood of attendance in cardiac rehabilitation [26], although overloading with too much detail in the acute stage can be counter-productive [27,28]. Importantly, with reducing length of hospital stays, there are diminishing opportunities for education and hence a variety of communication opportunities are important for sustained secondary prevention. The transition of care into general practice post discharge also represents an opportune time to sustain health education and support.

For secondary prevention of CHD, information exchange should be flexible and individually tailored based on patient risk factor profile, health needs and individual circumstances, such as socioeconomic background, gender, employment status, and geographical location [9,24,29]. Further, offering patients choice and actively engaging them in treatment decisions and giving choices have been shown to improve health outcomes [30,31,32]. An example is CHOICE, a brief, patient-centred intervention for ACS survivors not participating in cardiac rehabilitation [24]. Additionally, setting mutually agreed goals and offering individual choice enhance relevance and independence to change behaviour [33]. CHOICE patients participated in a modular program of an in-person clinic visit (1 h) followed by telephone support (for over 3 months), encompassing mandatory cholesterol lowering and tailored preferential risk modification. A significant improvement in modifiable risk profiles and risk factor knowledge of ACS survivors over 12 months was observed in the CHOICE participants. A modular concept is central to the patient-centred approach and allows patients to focus on problem areas [34]. The patient is therefore focused on addressing risk factors that are directly relevant to them, rather than expending time and effort increasing knowledge about all information needs perceived to be relevant for this population.

Another example of achieving coordinated secondary prevention of CHD for all in need is the Secondary Prevention for All in Need (SPAN) published framework [23]. SPAN enables a flexible but standardised approach and includes four core elements: (i) assessment, (ii) information sharing, (iii) individualised risk factor management, and (iv) ongoing support with follow-up. Each element provides flexible alternatives in terms of contact, setting, duration and format. For the information-sharing element, SPAN proposes that patients receive education about the pathophysiology of CHD, symptom management plan, importance of medication adherence, staying active, healthy eating and when to return to work. The information is ideally delivered using a range of resources including multi-media (e.g., slides, video) and other resources (e.g., flipcharts, leaflets and verified websites). The suggested educational content proposed by SPAN closely aligns with the evidence based Six Steps to Cardiac Recovery conversation guide (Six Steps guide) by the National Heart Foundation of Australia (NHFA) [35]. The Six Steps guide assists health professionals to facilitate information delivery at the patient’s bedside. This includes explaining the diagnosis, highlighting risk factors, encouraging patients to attend cardiac rehab, medication adherence, warning signs of heart attack, and the importance of following up with a GP and cardiologist [36].

This review was commissioned by the NHFA to assist with the iterative redesign of their patient support program offering assistance for people living with CHD. Specifically, given the importance of education and information-sharing for patients with CHD and the diverse nature of current offerings, the purpose of this review is to summarise the:i.Guiding principles, content, context and timing of information and education that is beneficial for supporting people with CHD;ii.Potential communication strategies including digital interventions.

## 2. Methods and Methods

This scoping review involved searching four databases (Web of Science, PubMed, CINAHL, Medline) for articles published from January 2000 to August 2022. The search was limited to English language. The following terms were used for title search: acute coronary syndrome, acute coronary event, acute cardiovascular event, coronary stent, CABG, coronary artery bypass graft, percutaneous coronary intervention, revascularisation, coronary artery disease, heart attack, myocardial infarction, angina, education, patient needs, patient information, post-discharge information, booklet, patient support, information needs, advise, counselling, digital intervention, digital strategy, text-message, SMS, web-content, smartphone application/app, tablet application/app.

Literature relevant to the review aims was identified through title and abstract screening by expert reviewers (C.Z. and J.R.). Grey literature was also searched using the reference lists of relevant articles found in the initial search and through the expert knowledge of the reviewers (C.Z. and J.R.). Relevant data was extracted and summarised. Evidence was subsequently synthesised according to the review aims. Points of commonality and difference within the literature were identified and combined to address the aims.

## 3. Results

### 3.1. Content and Timing of Education and Information

The content of education and information for patients with CHD can be varied and should be founded on several guiding principles outlined in Box 1.

Box 1Guiding Principles for patient information and education.Need for flexibility within and between peopleBe suitable for people from diverse and disadvantaged groups with consideration of language, culture, gender, socioeconomic status, health literacyBe evidenced-based and follow guidelinesConsider behaviour change concepts and goal-settingBe positive and meaningfulUse lay language, be systematic, and include visual presentation of materialsBe readily availableInvolve, and be suitable, for family and carers where appropriate

### 3.2. Education and Information Content

There is a variety of research reporting detail about content of educational strategies. A 2003 systematic review (14 studies) provided a thorough summary of the patient education and information needs for people who have had a myocardial infarction (MI) [37]. In terms of prioritisation, meta-analysis showed that information about risk factors (including treatment options [34]) ranked as most important overall, followed by cardiac anatomy and physiology, medication, and physical activity [37]. Miscellaneous items, diets and psychological factors ranked lower but were still considered important [37]. Studies not in the meta-analysis also highlighted the importance of dealing with emotions and stress, preventative prognostic, cognitive and affective information needs, as well as information for spouses. Table 1 summarises the content categories reported in the review by Scott et al., (2003) and includes additional, and more recent evidence to support the importance of each category. Studies also found that patients were enthusiastic to receive concise and positive information, such as capacity of the heart to heal and benefits of risk factor modification [38]. Another aspect, not mentioned in the content table, are studies that identified the need for information on factors associated with sexual activity [39,40,41].

The review by Scott et al., is limited to patients’ post-MI and is focused on inpatient and early discharge care. Therefore, it is also important to consider other areas of education content. These include but are not limited to strategies for role resumption, importance of regular primary care visits [23] and, for those who have undergone coronary artery bypass graft surgery, sternotomy care, physical activity restrictions and pain management are also important in the acute phase [55]. Behavioural considerations are also important in terms of education and information. For example, understanding how to set Specific, Measurable, Achievable, Relevant and Time-bound (SMART) goals and work in partnership with healthcare providers and carers in pursuit of life-long management and care. Goal-setting is an established strategy used in behaviour-modification programs in which patients set realistic and optimistic goals that assist in facilitation of changing health-related behaviours and long-term maintenance [56]. For patients, knowing how to set flexible, measurable, realistic, achievable, timely goals is the key to goal attainment regarding behavioural change, and consequently, decreasing the risk of further coronary episodes [57]. Delivering information about how to set SMART goals is feasible and practical to include in patient education and could easily be delivered by health professionals or through other communication means [58]. Furthermore, the review by Scott et al., was also published in 2003 and hence more contemporary considerations include the importance of flexibility related to individual circumstances, such as culture, gender, and socioeconomic considerations, among others. Contemporary studies also suggest that patients and their carers seek positive and practical information [38,59,60]. Patients have also expressed a need for information provided to be from a credible source or health professional [60].

### 3.3. Context

Having explored optimal content of information and education, it is then important to consider who, when and how such information should be provided. In several studies, CHD patients believed a general practitioner or physician should be the primary informant [37,38,54] and thought that clinicians can provide them with sufficient information, specifically regarding their individual diagnosis [61]. Other studies revealed that patients thought cardiac rehabilitation staff, cardiology specialists [54], or simply a professional contact who knows about the individual patient’s case/event [62] would be best placed for exchange of information. In one qualitative study, a participant also suggested an open telephone line for when problems arise [62]. Health professionals have been reported to express a desire to share more information [61], hence, this should be taken into consideration and used to full advantage. However, it is important that advice provided is evidence-based but also considers each patient’s perceived needs. Information exchange with a health professional also encourages shared decision making, which is also an important component of knowledge transfer and enhancement of the patient’s outcome [63].

### 3.4. Timing and Duration

Of the factors to consider for optimal information exchange (i.e., context, content, methods), it appears that the timing of the delivery is most variable. As well as when information is best delivered, it is also important to consider that it differs between and within individuals. For example, a patient’s ‘readiness’ for information is also variable and needs to be considered at multiple different timepoints in the course of their disease. A qualitative study reported that patients prefer less information during the emergency and in-hospital care stage [25]. In contrast, authors have suggested that introducing learning during the acute period may be of some importance [44], as uptake and adherence of secondary prevention behaviours could be time-sensitive [60,64]. Several articles proposed that patients have greater capacity for learning after discharge and in early recovery [38,61]. This may be due to their ability to cope with information and readiness to gain more control over their health.

In terms of duration, a systematic review of reviews found no consensus regarding number of information sessions, total contact hours or duration [65]. However, there was consistency between frequency of the educational session (either weekly or monthly). The average number of topics covered was 3.7 topics per education session [65]. Unsurprisingly, delivery of smaller packages of information over time was favoured by patients [46]. Ultimately, secondary prevention is lifelong and hence communication and support should be regular and ongoing. Longer-term communication can primarily be driven by at least annual primary care reviews [66].

### 3.5. Communication Strategies

There are increasingly varied ways in which information and education can be delivered. They include verbal, written, face-to-face, telephone and electronic communication. Regardless of which strategy or combination is used, it is critical to ensure the availability of concise, high quality, understandable health information, which is particularly important, considering the burden of CVD is greater for people with low health literacy [67].

#### 3.5.1. Face-to-Face

Verbal and face-to-face delivery is preferential for many patients [38]. Generally, these interactions are with a health professional (e.g., GP, cardiologist, cardiac rehab co-ordinator, nurse, allied health or Indigenous Health Worker). It is reasoned that a face-to-face approach enables the individual to ask questions, offers an opportunity to gauge truthfulness of the information being delivered and is seen as more personal than other methods [38]. Other advantages include a more dynamic and rich interaction with both verbal and non-verbal communication, rapport building with the health professional, and provides an opportunity to ask questions and clarify understanding. On the other hand, face-to-face interaction can be challenging for a variety of reasons, such as for people who live in remote areas, and where there are barriers, such as competing work or carer requirements, financial costs associated with travel, lack of transportation, restricted parking and inconvenient appointment times [68].

In Australia, eligible patients with CVD can access the Medicare Benefits Schedule (MBS)-funded Chronic Disease Management plans to support regular primary care review and establishment of management plans [69]. This scheme is designed to support systematic review and follow-up and can be used to assist patients to achieve regular care coordination and removes the need for making ad hoc consultations. A primary advantage is the opportunity to have regular face-to-face interactions with a health professional. Recent data suggests that half of Australians access chronic disease management plans with their GP within a year of being hospitalised for ACS.

#### 3.5.2. Written Material

Written materials also offer a valuable and potentially inexpensive strategy for reinforcing the verbal information [38]. Providing written information increases patients’ knowledge of disease [26,27,28] and reduces distress [29] and decisional conflict [30] because it reinforces verbal instructions and serves as a home resource [31]. Effective patient information must be evidence-based, clearly presented and most importantly, involve patients throughout the process of development [32,33,34]. Information materials should succinctly deal with the relevant messages [31], be written in the active voice with personalised messages and use simple terminology [14,31,35]. Layout should be clear, simple and consistent [14,35] and use subheadings to allow patients to efficiently sift through the information [17,31].

Examples of written materials that support people with CHD include the Heart Foundation’s My Heart, My Life program which includes two information booklets developed by the Heart Foundation [70] and the UK-developed Heart Manual [71]. Both are now also available in digitised versions. My Heart, My Life has been successfully piloted in Australia since 2019 (building upon a prior more text heavy version) and has shown positive patient/carer and health professional satisfaction. The Heart Manual was developed as a self-help package for patients after uncomplicated MI and is as effective as hospital-based rehabilitation for psychological, behavioural, biological, service and cost outcomes [72]. The manual provides simple explanations about CHD, secondary prevention and stress management, and types of physical activity intervention. Similarly, the My Heart, My Life booklet provides information about the anatomy and physiology of the heart, what to expect in hospital, recovery and behavioural change priorities and information about service and support. Both written materials are colourful, engaging and deliver simple messages [70,72].

#### 3.5.3. Visual Materials

Visual communication aids, such as images, videos and heart models, may be an effective delivery technique [38,73,74,75]. Patients prefer positive images that are colourful and that engage with text, as well as images of real people rather than cartoons [74]. Visual communication is also successfully used across digital platforms. For example, greater patient satisfaction was reported when receiving SMS text messages that included a visual aid [74] and tablet-delivered waiting room education increased motivation to improve cardiovascular health-related behaviours [75].

It is important to consider the availability of high-quality, clear and understandable information for people with CVD, as the burden of disease is greater for people with low health literacy. Additionally, some cardiac patients may feel unwell, be in pain, feel fatigued and/or be emotionally distressed during this time, which may acutely impact their health literacy [76]. A study of publicly available online CVD primary prevention decision aids (n = 25) found that the aids were understandable but only had moderate actionability and a high readability level beyond the literacy levels of the general population [67]. The study highlighted the need for more reliable tools for certifying decision aids to assess literature design issues, such as use of white space, images that are consistent with text and clear direction for next steps [67]. Additionally, a review of patient consent forms showed the need for visual aids to have clear titles or captions, be presented in a logical sequence, and use illustrations and photographs that are clear and uncluttered [76]. Thoughtfully designed visual aids could improve the written materials and understandability for people with a lower health literacy.

### 3.6. Digital Health Interventions

Digital health is an umbrella term covering technologies ranging from telehealth and remote monitoring, mobile management or monitoring apps, electronic data capture tools, big data predictions, artificial intelligence solutions, and decision support systems [77]. The recently published World Heart Federation Roadmap for Digital Health in Cardiology outlines the potential of interventions, such as text messaging, smartphone apps and wearable devices for patients with heart disease [35]. These interventions use computing platforms, connectivity, software, and sensors and offer an emerging frontier for preventing and managing CVD and range from wellness and prevention to prescription use as a medical device [35]. The COVID-19 pandemic has accelerated the prioritisation, development and implementation of such strategies [35]. Importantly, the World Health Organisation highlighted the value of patient and stakeholder engagement in the development of digital interventions [78]. Summarised below are several key digital health strategies that have been used to improve education and support for people with CVD.

#### 3.6.1. Telehealth

Provision of telehealth models using the phone and/or internet could potentially narrow the evidence practice gap and help to increase the uptake of secondary prevention for those who cannot or do not attend formal programs [79]. The use of telehealth escalated dramatically during the COVID-19 pandemic and can potentially improve access to CVD treatment by ensuring constant communication between patients and their health providers, and ensuring timely monitoring and remote counselling that support CVD management [35]. It is flexible, less threatening for the patients, cuts times and costs of clinic visits and also enables services to reach a large number of people, who may otherwise experience barriers when attending in-person appointments [79]. Evidence also shows that it is associated with lower re-hospitalisation [80] as well as being effective in CVD risk factor reduction [79], which could subsequently lead to fewer recurrent events.

Locally, two examples of programs that involve telephone consultations include CHOICE [32] and COACH [81]. CHOICE, which involves follow-up phone calls for three-months, showed that a patient-centred modular approach involving telehealth is both feasible and effective in reducing individual risk factors and improving coronary outcomes [82]. The COACH program, involving a dietician (with a background in education) who coaches CHD patients in behavioural change via telephone consults, has shown that telehealth can improve total cholesterol and LDL cholesterol levels in patients six months post-hospitalisation with CHD [81]. Both CHOICE and COACH are programs that support rather than replace usual medical care.

#### 3.6.2. Websites

With 91% of Australians now having access to the internet at home [83], websites can provide health information access, reaching consumers where they are looking for the information. Patients tend to prefer to use the internet for ease of access to information [84]. Other advantages of using websites for information delivery are that they are also highly scalable and reach a diverse population [85]. Disadvantages include the risk that information can be generic, of poor quality and not a priority for most patients and that it may require a high health literacy level to understand [86].

Review studies report that patients with chronic diseases, such as CHD, are seeking information from credible sources [84]. Credibility is evaluated based on factors, such as authority and credibility of authors, recommendations from others and references [87], and include stakeholders, such as the Heart Foundation. Studies have also found that patients still trust that their GP can provide more clinical expertise and experience [84], so websites should be used in combination with regularly seeking information from primary care providers. Evidence about the effectiveness of websites in providing secondary prevention information for this population group is minimal. Two randomised control trials (RCTs) have shown that a web-based app linked to primary care electronic health records has borderline improvements in risk factors and modest behavioural changes in people with CVD [88,89].

#### 3.6.3. Text-Messaging Programs

The most researched digital health interventions are text message programs, in particular for people with chronic disease. This is likely because they are a convenient and inexpensive strategy that can be fully automated with communication directly to a patient’s mobile phone. As patients do not require a smartphone internet access, scalability of these programs is enhanced. Text message programs can improve single risk factors, including tobacco smoking [90], high blood pressure [91], physical activity [92], overweight [93], and medication adherence [94,95]. As a result, text message programs for smoking cessation and medication adherence have been implemented. An example is the SmokefreeTXT program in the US [96].

The TEXTME RCT (n = 710 patients with CHD) tested a 6-month intervention addressing multiple cardiovascular risk behaviours and found significant improvements in total and LDL-cholesterol, blood pressure, body mass index, smoking rates, physical activity and adherence to dietary guidelines [97]. In TEXTME, four semi-personalised messages were sent automatically on random days and times during the week [97]. Text message content was co-designed with patients and clinicians and based on behavioural psychology [59]. The TEXTME program was also found to be engaging and useful for patients [60] and cost-effective [98]. The program won the 2016 Google Impact Challenge and has since been implemented by the Heart Foundation within the My Heart, My Life program.

#### 3.6.4. Smartphone Apps

Escalation of smartphone and ‘tablet’ ownership has resulted in the rapid growth of digital health interventions delivered via these new devices. These devices can support apps, which are computer-like programs with many functionalities [99]. Apps can support patient education, store health information, enable electronic storage of health data, automated reminders, monitor and manage health conditions and more. A major advantage is that they can overcome access barriers, such as distance and geography [100]. App usage is very common among young populations, however, usage and ehealth literacy may be variable among some older populations [100].

A review of smartphone apps to prevent CVD highlights the importance of simple, credible and evidence-based information based on behaviour change concepts, real-time data tracking, virtual positive reinforcement, app personalisation, social elements, and ensuring privacy [101]. A 2018 systematic review found app-users had reduced rehospitalisation rates and improved patient knowledge, quality of life, psychosocial well-being, blood pressure, body mass index, waist circumference, cholesterol, and exercise capacity [64]. However, these reviews acknowledge that evidence remains limited and more large trials with objective outcomes are required [64,99,101].

Examples of apps for which there have been RCTs include the CONNECT study, HERB Digital Hypertension 1 (HERB-DH1) and MEDication reminder APPs to improve medication adherence in CHD (MEDApp-CHD) studies [77,89,102]. CONNECT evaluated a consumer-focused and interactive web app integrated with each patient’s primary care electronic health record [89]. The study found modest improvements in risk factors and lifestyle behaviour; however, it was not significant [89]. The HERB-DH1 RCT (n = 390) found an interactive smartphone app retrieving home BP monitoring data to generate a personalised program of lifestyle modifications and improved ambulatory, home and office systolic BP in Japanese patients [102]. The MEDapp-CHD RCT showed that patients with CHD who used medication reminder apps had better medication adherence compared with usual care [77]. MEDApp-CHD also found that basic (in this case Heart Foundation’s My Heart, My Life app) and advanced (complex and interactive features) apps had similar effects on medication adherence and 81% of participants found the apps useful, easy to use and helpful for reminding them to take their medications [77]. In a systematic review [103], advanced apps, for example the Medisafe app, that have more complex features (highly interactive, customisable, high level of visual appeal) help to add to interest and increase engagement of the user [77,103].

#### 3.6.5. Wearable Devices

Wearable devices can collect information, perform data processing and summarise relevant information by connecting with another device, such as a smartphone app or a Bluetooth device [104]. Such devices have become extremely popular and offer continuous data monitoring of vital signs, activities and behaviours [100]. An example of wearable devices for CVD management is physical activity trackers that can be worn on the wrist or built into smartphones, used to provide information on daily step counts, distance walked, energy expenditure and heart rate [99]. A 2019 systematic review (28 studies) investigated the effects of wearable activity trackers and found a significant increase in daily step count, physical activity, and energy expenditure among the intervention groups compared to controls [105]. Recently, the Apple Heart Study showed how wearable devices can be used to detect heart rate irregularity and atrial fibrillation [106]. In the study, each participant’s (n = 419,297) wearable device detected an irregular pulse via an algorithm. The detection of an irregularity prompted a telehealth visit and full electrocardiography (ECG). Results found 0.52% of participants had an irregular pulse identified, and one-third of those who returned an ECG had atrial fibrillation. The Apple Heart study provides an example of a virtual- and practical- study design where participants’ own wearable devices assessed outcomes, even though there was a low rate of irregular pulse notification.

## 4. Discussion

Improving post-discharge care through secondary prevention strategies (healthy living, adherence to medicines) is a current national and international priority. This review highlights that information-sharing, decision-making, goal-setting, positivity and practicality are important aspects of secondary prevention and should be patient-centred and evidenced based with consideration of patient need and preference.

Reassuringly, people with heart disease are seeking information from credible sources and that includes health professionals, such as cardiologists, GPs, cardiac rehabilitation staff, nursing and allied health providers or from credible organisations. Design of information resources for people with CHD should involve people who have an awareness of the value of patient education and health literacy. Initiation and duration of education is highly variable between and within people with CHD, but secondary prevention is lifelong and so communication and support should be regular and ongoing. For individuals after a cardiac event, important information content includes information about CVD risk factors and how to manage them, understanding heart disease, warning signs, medications, psychosocial considerations, and role resumption. There are increasingly varied ways in which information and education can be delivered and they include verbal, written, face-to-face, telephone and electronic. Importantly, written materials should be consistent, easy to understand, written in plain language, non-judgemental and suitable for people with low health literacy. Visual materials, such as images, videos and physical models, can be helpful for communication of some concepts, including pathophysiology, risk factors and practical strategies.

Text messaging programs, smartphone applications and wearable devices are some examples of digital health strategies that can facilitate education and support for patients with heart disease. Ideally organisations wishing to develop patient education resources should consider flexibility, variety and patient-centredness.. Figure 1 provides a schematic overview of the education and information requirements for patients if we are to optimise post-discharge care through secondary prevention strategies (healthy living, adherence to medicines), which is currently a national and international priority. Most importantly, the education and information should be patient-centred, and there is no one size fits all approach that suits all patients at all stages, hence flexibility and a suite of resources and strategies that is flexible and adaptable to patient needs is optimal.

In terms of a future perspective, flexibility is likely to be the key with a fundamental focus on core principles and requirements. Society is ever-change and so delivery modes will need to vary, however the underlying purpose and intent can remain consistent with a focus on evidence-based and contemporary content. Another important area of importance is health literacy and e-health literacy where health information should be universally available and meaningful to all people no matter where they are from and what their background may be. Technology may be helpful for some, while others may prefer in-person communication. Whichever the format and delivery mode, the primary purpose should remain supporting each person to optimize their cardiovascular health.

## 5. Conclusions

In conclusion, a critical aspect of coronary heart disease (CHD) care and secondary prevention is ensuring patients have access to evidence-based information and that they engage with it. This review highlights that the importance of information in decision-making, goal-setting, positivity and practicality are important aspects of secondary prevention and should be patient-centred and evidenced based with consideration of patient needs and preferences. However, initiation and duration of education is highly variable between and within people, hence communication and support should be regular and ongoing. Text messaging programs, smartphone applications and wearable devices are some examples of digital health strategies that can facilitate education and support for patients with heart disease. Importantly, there is no one size fits all approach that suits all patients at all stages, thus flexibility and a suite of resources and strategies is optimal.

## Figures and Tables

**Figure 1 ijerph-20-01723-f001:**
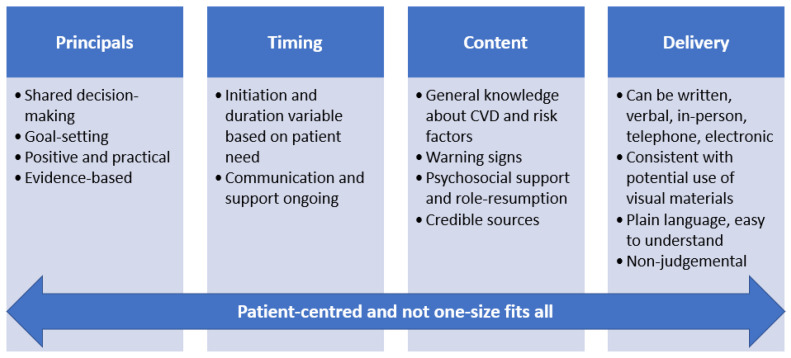
Schematic overview and education and information needs for patients.

**Table 1 ijerph-20-01723-t001:** Reported information needs for patients’ post-Myocardial Infarction (based on review findings of Scott and Thompson 2003 [37]).

Information Category	Content
Medication [38,42,43,44,45,46,47,48,49,50,51,52]	List of medication are prescribed and being taken Dosage or regimen Indications and side effects of each medication
Anatomy and physiology of the heart [38,43,44,47,49,52,53]	Basic structure and function of the heart Explanation of causes and pathophysiology of a heart attack, atherosclerosis, and angina
Symptom management [38,48,49,50,51,52,53]	Signs and symptoms of complications requiring medical attention Managing chest pain and instructions for medication for angina Learning and understanding warning signs of a heart attack and appreciating that these are variable
Pathology and impact of disease [38,46,50,54]	What happened to the heart during the cardiac event (culprit vessel) Knowing the chances of, and how to recognise, another heart attack Knowing what the diagnosis of heart-attack means Understanding treatment options
Risk factors and cardiac rehabilitation [43,44,47,53]	Understanding of clinical and lifestyle risk factors (modifiable and non-modifiable risk factors) Importance of regular assessment of and modifying risk factors, such as smoking, diet, physical activity General understanding of blood cholesterol levels, blood pressure and body weight Support and understanding about quitting smoking and reducing alcohol intake (where relevant) Undferstanding what constitutes a good diet and how diet can be modified with healthy eating tips Importance of regular physical activity, reducing sedentary time and goal-setting Understanding the value and availability of cardiac rehabilitation
Psychosocial factors [43,49]	Understanding emotional reactions including stress, anxiety, depression Basic strategies for expressing and coping with emotional reactions, referral pathways Relating to family and friends
Wound care [42,45]	If relevant, basic information about how to care for any wounds and signs of complications or poor healing Exercises for patients who have undergone median sternotomy during cardiac surgery [55]

## Data Availability

Data is available upon request.

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
