# Peer review of "Information Needs and Communication Strategies for People with Coronary Heart Disease: A Scoping Review"

_ijerph, 2023, doi:10.3390/ijerph20031723_

Round 1

Reviewer 1 Report

To improve post-discharge care of coronary heart disease (CHD), the authors in this review summarized guidelines and strategies that actively engage CHD patients in improved health behavior to slow or reverse progression of disease.

The authors searched four databases (Web of Science, PubMed, 129 CINAHL, Medline) for articles published from January 2000 to August 2022 to identify relevant literatures. Based on these literatures, the authors summarized guiding principles, education and information content, timing and duration, communication strategies including face-to-face, written material, visual materials, digital health interventions(telehealth, websites, text-messaging programs, smartphone Apps, wearable devices). The authors concluded that

patient-centered approaches to actively engaging them in CHD care and secondary prevention is critical to improve post-discharge recovery.

This review is clear, well organized, comprehensive and of relevance to the field. It also cited relevant references. The statements and conclusions drawn coherent and supported by the listed citations.

Table1 can be further improved. Currently, it is not easy to correlate content(right column in the table) to its information category(left column in the table). It will be easier to follow if the authors can highlight different information category and its corresponding content with distinct color in the table.

Author Response

  1. Table1 can be further improved. Currently, it is not easy to correlate content (right column in the table) to its information category (left column in the table). It will be easier to follow if the authors can highlight different information category and its corresponding content with distinct color in the table.

 Response: We think this is large a formatting issue and have left justified the table and used a light shade to distinguish the different rows. We understand the editorial staff will need to determine whether the shading is in line with standard journal formatting and whether a light colour shade can be used.

  1. English language and style are fine/minor spell check required

Response: Reviewed and checked, we have made a number of corrections. The remaining spelling errors are highlighted because we have used English rather than American spelling. We are unsure which is required for the journal.

Reviewer 2 Report

This review, that is commissioned by the National Heart Foundation Australia, provides an overview on guiding principles, content, context and timing of information and education for supporting patients with coronary heart disease. The topic is interesting and relevant and overall the review is well written. The number of tables/figures is limited in this review, and although this is a narrative review, I do think that the readability of this manuscript would improve when the information is also presented in some nice figures and/or tables. I would suggest to add at least an illustration that summarizes the main findings of this review. I would also suggest to add on a section “ Future perspectives” where the authors outline what in their opinion future research should focus on. Please see my other suggestions below.

Comment 1: The first paragraph is pretty much focused on the Australian population. I do appreciate that this is a review that was commissioned by the National Heart Foundation Australia, but I would like to suggest to introduce the topic more broadly.

Comment 2: Two examples are provided in the introduction section (CHOICE and SPAN). Please consider to remove these examples from the introduction section and add it to the results section instead.

Comment 3: Please spell out the abbreviation NHFA in the introduction section.  

Comment 4: Search. I would like to suggest to add on a table with the specific terms for each database how the search has been done (Mesh terms etc.). If available, also present a flow diagram with the number of articles identified, screened and selected.

Comment 5: Table 1. Why did the authors decide only to present a table from a review that has been published in 2003? Please provide a summary table/figure where an overview is presented that also includes contemporary findings.

Author Response

The number of tables/figures is limited in this review, and although this is a narrative review, I do think that the readability of this manuscript would improve when the information is also presented in some nice figures and/or tables. I would suggest to add at least an illustration that summarizes the main findings of this review. I would also suggest to add on a section “ Future perspectives” where the authors outline what in their opinion future research should focus on.

Response: We have added a new figure as suggested that provides an overview of the education and information needs. We are not certain of the journal formatting requirements so can adjust the figure as advised by the editorial team. We have also added an additional paragraph that is focused on future perspectives where we highlight the importance of flexibility and health literacy.

  1. The first paragraph is pretty much focused on the Australian population. I do appreciate that this is a review that was commissioned by the National Heart Foundation Australia, but I would like to suggest to introduce the topic more broadly.

Response: We have refocused the opening of the Introduction with global statistics as suggested and changed reference #2 from one that represents Australian to international data.

  1. Two examples are provided in the introduction section (CHOICE and SPAN). Please consider to remove these examples from the introduction section and add it to the results section instead.

Response: We feel these are 2 overarching models that provide background information and and hence do not fit directly in the Results as a clean switch and as such we would prefer to leave these as they are.

  1. Please spell out the abbreviation NHFA in the introduction section.  

Response: Spelled out at first use as suggested.

  1. I would like to suggest to add on a table with the specific terms for each database how the search has been done (Mesh terms etc.). If available, also present a flow diagram with the number of articles identified, screened and selected.

Response: We think this is a nice suggestion but given the narrative nature and with the addition of a new diagram and changing of Table 1 we do not feel further additional tables and figures are necessary.

  1. Table 1. Why did the authors decide only to present a table from a review that has been published in 2003? Please provide a summary table/figure where an overview is presented that also includes contemporary findings.

Response: The purpose was to present information about educational material, we were unable to identify much more contemporary frameworks research on this specific topic and hence with the 2003 one. We have now included a new figure as per Reviewer 1’s suggestion.